# Assessing pain severity and treatment outcomes in patients with low back pain: A Structural equation modeling approach at the center for the rehabilitation of the Paralysed, Bangladesh

**Mohammad Arifur Rahman, Sanjida Tasnim** *, **Md. Forhad Hossain**

Department of Statistics and Data Science, Jahangirnagar University, Savar, Dhaka, Bangladesh

* sanjidatasnim@juniv.edu

## Abstract

### Objectives

This study aims to observe the associated risk factors of lower back pain and the factors that increase the pain severity. So, the main objective of this research is to identify the factors which may cause the lower back pain and the causal effect on the pain severity and respective treatment. This study also tries to determine the demographical characteristics of the low back pain patients and determine the inter relationship of psychological health, work stress and treatment effect with the pain disability index.

### Study design

In this cross-sectional study, 200 patients with lower back pain were interviewed who were taking treatments from the physiotherapy department at the Center for the Rehabilitation of the Paralysed, Savar, Dhaka, Bangladesh.

### Methods

A quantitative research model has been performed to observe the relationship between different causes of low back pain effects on the patients. Different statistical analysis including structural equation modeling have been performed to observe their pain severity and treatment effect.

### Results

The study found 64% (128) of the total participants as male and 36% (72) as female among 200 patients of low back pain. The study also observed the highest portion of the patients belong to the age group 39 to 45 years (21.5%). On the basis of BMI, obese weight respondents were 26.5% (53), overweight respondents were 37% (74), normal weight respondents were 33% (66), and underweight respondents were only 3.5% (7). Here, sex, body mass index (BMI), living place and educational status have significant association with pain

**Data Availability Statement:** All relevant data are within the manuscript and its Supporting Information files.

**Funding:** The authors received no specific funding for this work.

**Competing interests:** The authors have declared that no competing interests exist.

disability index (PDI). On the other hand, smoking tendency of patients has insignificant relationship (p>0.05) with pain disability index (PDI). The path coefficients of the structural equation model identified that all the null hypotheses of no significant relationship have been rejected for 5% level of significance. The hypothesis of psychological health is positively related to pain severity of a patient has an acceptable strength ($\beta$ = 0.745, p<0.001) and a positive direction. Another hypothesis (Psychological health is positively related to the treatment of a patient) shows an acceptable strength ($\beta$ = 0.401, p <0.001) and a positive direction. Work stress is also found to be positively related to pain severity of a patient with an acceptable strength ($\beta$ = 0.544, p < 0.001) and a positive direction. The hypothesis (Work stress is positively related to the treatment of a patient) has an acceptable strength ($\beta$ = 0.322, p< 0.05) and a positive direction. The hypothesis (pain severity is positively related to the treatment of patients) shows an acceptable strength ($\beta$ = 0.801, p < 0.001) and a positive direction.

## Conclusion

The research found out the psychological health situation and work stress of patients are significantly related with pain severity with acceptable strength. Also, Pain severity is significantly associated with treatment scheme intensity.

## 1.Introduction

Low back pain (LBP) is one of the most common health problems throughout the world. Almost 80% of the population experience LBP at least once in their lifetime. The low back or the lumber area of a human body serves as structural support, movements and protection of body tissues. When there happens an injury such as muscle sprains or strains due to certain diseases or due to poor body mechanics for sudden movements, subjects experience mild and severe pain which can last from few days to few years. It is a major health problem with two third of adults suffering from LBP at some time in their lives and almost 12% to 44% have LBP at any particular time [1]. There is evidence that 12% to 26% of children and adolescents experience low back pain although most cases of low back pain occur in persons between the ages of 25 and 60 years, peaking at about 40years [2].

The chance of developing lower back pain is increased by certain demographic characteristics, such as age, gender, and others, as well as by risk factors for LBP, such as weight lifting, using vibrating equipment, leading a sedentary lifestyle, obesity, smoking, having an increased lumbar lordosis, scoliosis, cardiovascular disorders, low socioeconomic level, and so on [3].

It is concerning that more and more people in Bangladesh are reporting LBP. Many hospitals are offering these patients various kind of physiotherapy treatments in an effort to lessen their pain. The purpose of this study is to determine if LBP treatment can lessen patients' pain intensity while also monitoring latent variables including socioeconomic status, work-related stress, patients' physical and mental health, etc. For exploring the intricate relationship between low back pain severity, professional stress, psychological health and treatment outcomes in patients, this study employs Structural Equation Modeling (SEM) approach. Through rigorous analysis, it aims to discover the interplay of these factors, shedding light on how they influence each other and ultimately patients' well-being. By exploring these complex connections, this research strives to enhance more effective interventions for physiotherapy within the context of Bangladesh's rehabilitation landscape.

## 2. Methods and materials

### 2.1 Study design

The cross-sectional analytical study was done in the Physiotherapy department at the Center for the Rehabilitation of the Paralysed (CRP), Savar, Dhaka, Bangladesh. A random sample of 200 patients were interviewed who had fulfilled the enrollment criteria attending the Physiotherapy department, Savar, Dhaka from February 2021 to May 2021. Some demographic attributes of patients have been included in questionnaire along with pain severity, work condition, work pressure, work environment, psychological health condition, problems faced in doing daily life activities, pain disability index, and suffering pain from how many days. Data was approved by the Center for the Rehabilitation of the Paralyzed (CRP), Savar, and Dhaka, Bangladesh. All the study subjects knew about the aim of the study and informed written consent was taken before collecting data.

### 2.2 Ethics statement

The study was approved by the Centre for Rehabilitation of the Paralysed Ethics Committee (CRP-EC), Savar, Dhaka, Bangladesh. The approval number is CRP-R&E-0401-239 and the date of approval is 26.01.2021. All study subjects were informed about the aim of the study and informed written consent was taken before collecting their personal data.

### 2.3 Questionnaire setting

The subjects were interviewed based on a well-set questionnaire. The questionnaire is comprised of five major parts. Part I and II contains personal and socio-demographic information about the subjects. Part III was developed based on the Dallas pain questionnaire for evaluating subject's initial and discharge cognitions about the percentage that chronic pain affects four aspects of patient's lives: (1) daily activities including pain and intensity, personal care, lifting, walking, sitting, standing and sleeping; (2) work and leisure activities including social life, travelling and vocational; (3) anxiety-depression and (4) social interest that includes interpersonal relationship, social support and punishing responses. Each question to measure Dallas pain index has its own continuous visual scale so that subjects can response more accurately about their situations.

Part IV was developed according to Oswestry low back pain disability index which is used to measure the patients pain disability index (PDI). Finally, treatment-oriented questions are established by the physiotherapist based on the regular treatment protocol in part V. Here each subject is assessed with an initial and outcome level of pain intensity after receiving certain doses of specific or need based treatment which the physiotherapist suggested for him/her.

### 2.4 General data collection

Personal information, socio-demographic information, initial and discharge question, and low back pain disability index data were collected through a separate questionnaire from the selected subjects within a fixed time stamp of 01.02.2021 to 03.05.2021.

### 2.5 Sample collection

The sample was taken while the patients with low back pain came into the physiotherapy department and got treatment for a minimum of 4 days from CRP. (Inclusion criterion): Nature of Low Back Pain, Functional Impairment, Age Range, Willingness to Participate,

Mechanical low back pain, Pain severity, Treatment, (Exclusion criterion): Lower back pain with other pain, with some injury.

## 2.6 Data analysis

The study aims to find the relationship between treatment and severity of pain in lower back part of patients along with their biological, demographic attribute and working environment. To focus on this part, structural equation modeling has been employed. Additionally, the association between several attributes have also been checked using Chi square test. All data are analyzed by statistical package for social science (SPSS) and SPSS AMOS.

**2.6.1 Structural equation modeling.** Structural equation model (SEM) is a multivariate technique designed with two-part, i) measurement model and ii) structural model. In measurement model, the relationship between the latent variable and observed variable is perceived and on the other hand structural model is used to investigate the loadings and estimating indicators. Construction of SEM includes the following steps.

*2.6.1.1 Specify the measurement model (CFA).* Confirmatory factor analysis (CFA) is a theory driven methodology that confirms a hypothesis. As a result, the theoretical relationship between the observed and unobserved parameters directs the analysis plan [4]. After defining the relationships between latent variables and their observed indicators, factor loadings, variances, and covariances are estimated.

*2.6.1.2 Specify the structural model.* This model is extended to include structural relationships between latent variables. Paths or regression coefficients between latent variables and any direct effects on observed variables are specified and error terms for the observed variables is included.

*2.6.1.3 Estimate model parameters.* SEM analysis is performed to estimate the parameters of the model. Factor loadings, regression coefficients, variances, covariances, and other parameters are determined based on the specified model.

*2.6.1.4 Assess model fit.* Fit of SEM model to the data is evaluated using fit indices. Common fit indices include Chi-square ($\chi^2$), comparative fit index (CFI), increment fit index (IFI), Tucker-Lewis's index (TLI), parsimonious comparative fit index (PCFI), parsimonious normed fit index (PNFI), root mean square error of approximation (RMSEA), and standardized root mean square residual (SRMR). Model's goodness-of-fit is assessed relative to established thresholds or benchmarks.

This Table 1 provides short labels and descriptions for various aspects related to psychological health pain severity, work stress and treatment protocols. It covers different dimensions of pain, including intensity, stiffness, depression, and fear of panic attacks. Additionally, it addresses issues related to work, such as problems, changes, and pressure in the workplace. Finally, it outlines treatment protocols for different levels of pain intensity over a four-day period, offering a comprehensive overview of factors relevant to pain management and work-related stress.

# 3 Results and discussions

## 3.1 Univariate analysis

Demographic characteristics are summarized in Table 2. In this research, a total of 200 patients have participated. Among them, 128 (64%) participants are males and 72 (36%) are females. The study found 21.5% of its respondents came from the age group 39–45 and 20% came from age group 53–59. Occupationally, 32.5% of respondents are housewives and 19.5% are service holder. The study observed 37% (74) of the patients are overweight whereas 26.5% (53) are obese and 33% (66) are normal in weight.

**Table 1. The excerpts of the questionnaire used in the research.**

| Short labels for the individual items | Description | Short labels for the individual items | Description |
|---|---|---|---|
| PH1 | Pain intensity | PS2 | Pain in stiffness |
| | Range: 1 to 4, no pain to severely pain | | Range: 1 to 4, no stiffness to severe stiffness |
| PH2 | Feel depressed | PS3 | Pain in twisting |
| | Range: 1 to 4, never to always feel depressed | | Range: 1 to 4, no problem to severe problem in twisting |
| PH3 | Fear of experiencing panic attack | PS4 | Problem lying on the bed |
| | Range: 1 to 4, fear to panic of experiencing an attack | | Range: 1 to 4, no problem to severe problem lying on bed |
| WS1 | Problem with doing work | TRT1 | First-date treatment protocol |
| | Range: 1 to 4, no problem to severely problem in doing work | | Range: 1 to 3, higher dose of treatment protocol for slight pain to severe pain |
| WS2 | Changes in the workplace | TRT2 | Second-day treatment protocol |
| | Range: 1 to 4, no changed to several change of work place | | Range 1 to 3, less high dose of treatment protocol than TRT1 for slight pain to sever pain |
| WS3 | Pressure in work | TRT3 | Third-day treatment protocol |
| | Range: 1 to 4, no pressure to too pressure inwork | | Range: 1 to 3, less high dose of treatment protocol than TRT2 for slight pain to severe pain |
| PS1 | pressure in bending | TRT4 | Fourth-day treatment protocol |
| | Range: 1 to 4, no pain to severe pain in bending | | Range: 1 to 3, less high dose of treatment protocol than TRT3 or same for slight pain to severe pain |

*PS = Psychological health, WS = Work stress, PS = Pain severity and TRT = Treatment

In the terms of family size and living place both are categorized into two groups i) small family (47.5%) ii) large family (52.5%) and i) urban area (41%) ii) rural area (59%). The pie chart (Fig 1) shows, the maximum number of respondents are affected by slight pain 97 (48.5%) in terms of the pain disability index, whereas 41% (82) feels moderate pain and 11% (21) severe pain in their low back area.

### 3.2 Bivariate analysis

The individual association of pain disability index with some demographical characteristics have been illustrated in Table 3. Though demographical factors are not the main contributor to pain severity, sex, BMI, education and living area showed statistically significant association with PDI. To measure pain severity, other multivariate models are established in later part.

From the bivariate table it is evident that, sex, BMI, living place and educational status have significant association with pain disability index (PDI), on the other hand smoking has no significant association ($p > 0.05$) with pain disability index.

### 3.3 Structural equation modeling procedure

**3.3.1 Hypothesis statement.** The hypotheses for the study are set as follows;

$H_1$: There is a significant relationship between psychological health and pain severity, $H_2$: There is a significant relationship between psychological health and treatment, $H_3$: There is a significant relationship between work stress and pain severity, $H_4$: There is a significant relationship between work stress and treatment, $H_5$: There is a significant relationship between pain severity and treatment.

**Table 2. Demographic attributes of the respondents.**

| Variables | Category | Frequency (percent %) |
|---|---|---|
| Age | 18–24 | 5(2.5%) |
| | 25–31 | 17 (8.5%) |
| | 32–38 | 23 (11.5%) |
| | 39–45 | 43 (21.5%) |
| | 46–52 | 29 (14.5%) |
| | 53–59 | 40 (20%) |
| | 60–66 | 25(12.5%) |
| | 67–73 | 13(6.5%) |
| | 74–80 | 2 (1%) |
| | 81–87 | 3 (1.5%) |
| Sex | Male | 128(64%) |
| | Female | 72(36%) |
| Marital status | Married | 174(87.0%) |
| | Unmarried | 21(10.5%) |
| Widow | | 3(1.5%) |
| | Divorced | 2(1.0%) |
| Family size | Small Family | 95(47.5%) |
| | Large Family | 105(52.5%) |
| Living place | Urban | 82(41.0%) |
| | Rural | 118(59.0%) |
| Occupation | Farmer | 15(7.5%) |
| | Day labor | 6(3.0%) |
| | Service holder | 39(19.5%) |
| Garments worker | | 12(6.0%) |
| | Driver | 8(4.0%) |
| | Businessman | 20(10.0%) |
| | Unemployment | 14(7.0%) |
| | Housewife | 65(32.5%) |
| | Teacher | 5(2.5%) |
| | Student | 7(3.5%) |
| | Others | 9(4.5%) |
| BMI | Underweight | 7(3.5%) |
| | Normal weight | 66(33.0%) |
| | Overweight | 74(37.0%) |
| | Obese weight | 53(26.5%) |

**3.3.2 Measurement model.** A measurement model (Fig 2) was developed including 4 latent constructs to investigate the significant factor for treatment for low back pain patients. CFA model showed that the four latent constructs in this research are psychological health, work stress, pain severity and treatment. PH1-PH3, WS1-WS3, PS1-PS4, and TRT1-TRT4 represent the observed (independent) variables and the association with them (factor loadings) indicates the relationship between unobserved (latent construct). To get construct reliability, validity and a better model fit, item with lower factor loadings (less than 0.5) were cut off from this model [5]. Also, cross-loading (Table 4) was determined to see the association between one latent variable and another observed variable.

In Fig 2, the strong correlation between psychological health and pain severity suggests that addressing psychological well-being is crucial in managing and understanding pain levels.

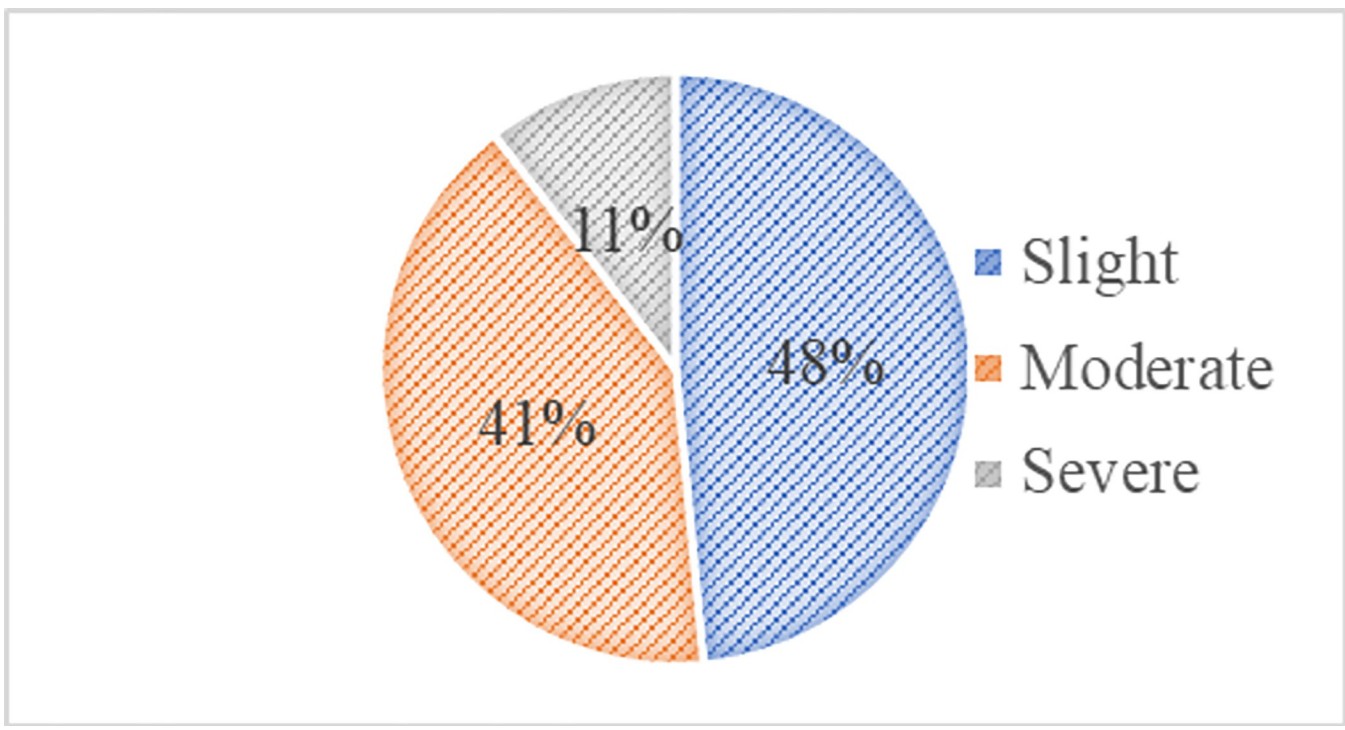

**Fig 1. Distribution of subjects according to pain disability index (PDI).**

Moreover, work stress is identified as a contributing factor to pain severity, indirectly influencing psychological health.

Here 14 indicators of the conceptual framework model were run with the help of AMOS software and the framework used in the hypothesis testing parts is illustrated in Fig 3. It should

**Table 3. Bivariate association of PDI according to other variables.**

| Variable | | Pain disability index (PDI) | | | |
|---|---|---|---|---|---|
| | | Slight | Moderate | Severe | p-value of $\chi^2$ |
| Sex | Male | 71(62.1) | 47(52.5) | 10(13.4) | 0.022** |
| | Female | 26(34.9) | 35(29.5) | 11(7.6) | |
| BMI | Underweight | 2(3.4) | 5(2.9) | 0(.7) | 0.019** |
| | Normal weight | 34(32) | 27(27.1) | 5(6.9) | |
| | Overweight | 30(35.9) | 32(30.3) | 12(7.8) | |
| | Obese | 31(25.7) | 18(21.7) | 4(5.6) | |
| Living place | Urban | 44(39.8) | 36(33.6) | 2(8.6) | 0.008 ** |
| | Rural | 53(57.2) | 46(48.4) | 19(12.4) | |
| Smoking | Yes | 31(26.2) | 18(22.1) | 5(5.7) | 0.304 |
| | No | 66(70.8) | 64(59.9) | 16(15.3) | |
| Educational level | Illiterate | 5(6.3) | 5(5.3) | 3(1.4) | 0.020 ** |
| | Primary | 17(26.7) | 29(22.6) | 9(5.8) | |
| | Secondary | 39(37.8) | 32(32) | 7(8.2) | |
| | HSc passed | 21(15) | 8(12.7) | 2(3.3) | |
| | Graduate & masters | 15(11.2) | 8(9.4) | 0(2.4) | |

$P < 0.05$**, $P < 0.001$***

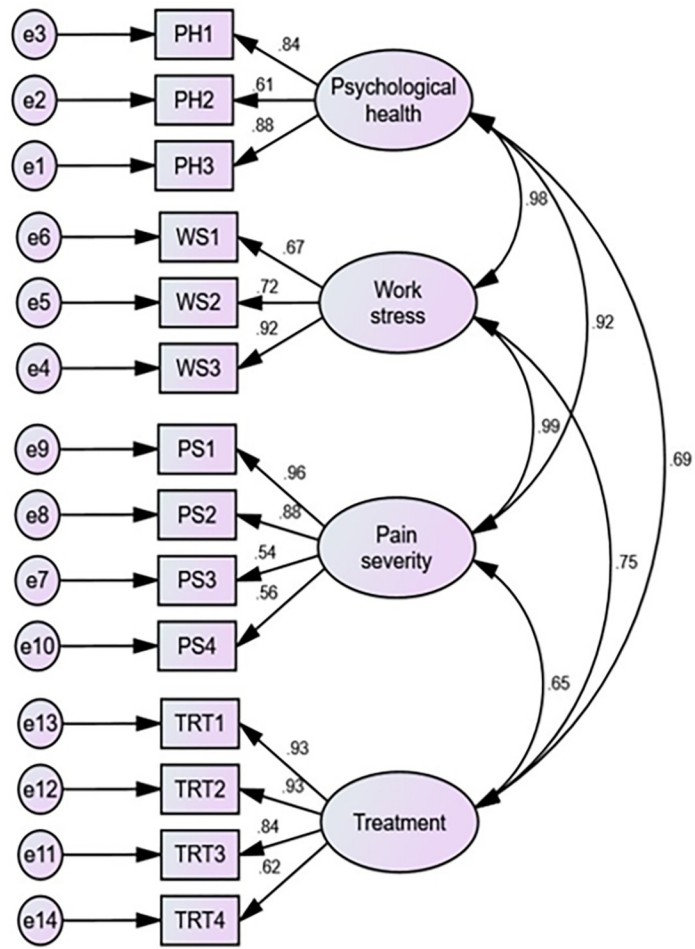

**Fig 2. Measurement model.**

**Table 4. Cross-loading.**

| Cross loading | | | | | |
|---|---|---|---|---|---|
| Construct | Item | PH | WS | PS | TRT |
| Psychological health | PH1 | 0.84 | 0.48 | 0.39 | 0.092 |
| | PH2 | 0.61 | 0.35 | 0.41 | -0.08 |
| | PH3 | 0.88 | 0.43 | 0.46 | -0.04 |
| Work stress | WS1 | 0.31 | 0.67 | 0.65 | -0.069 |
| | WS2 | 0.40 | 0.72 | 0.43 | -0.075 |
| | WS3 | 0.37 | 0.92 | 0.61 | -0.088 |
| Pain severity | PS1 | 0.39 | 0.59 | 0.96 | 0.015 |
| | PS2 | 0.78 | 0.49 | 0.88 | 0.032 |
| | PS3 | 0.67 | 0.57 | 0.54 | 0.019 |
| | PS4 | 0.23 | 0.38 | 0.56 | 0.032 |
| Treatment | TRT1 | -0.065 | 0.019 | 0.07 | 0.93 |
| | TRT2 | 0.074 | -0.078 | -0.19 | 0.93 |
| | TRT3 | 0.069 | 0.047 | -0.12 | 0.84 |
| | TRT4 | -0.045 | 0.015 | 0.068 | 0.62 |

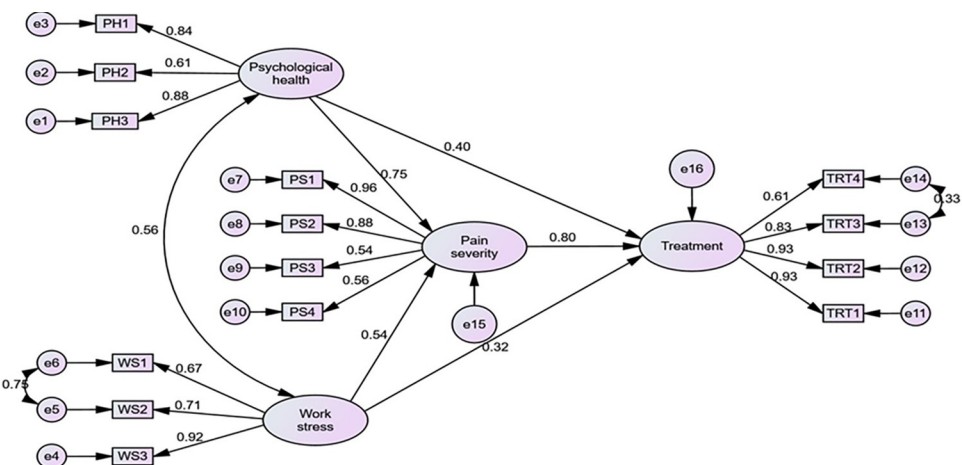

**Fig 3. Structural equation model.**

be noted that psychological health constructs had three indicators, work stress had three indicators, pain severity had four indicators, and final constructs treatment had four indicators. Also, in Table 4, cross-loading shows the result of the correlation (loading) of one construct with other constructed items. It seems that every latent construct is strongly loaded with familiar items. Before describing the final SEM model, the measurement model was run to test the reliability and validity (Table 5) of those indicators.

In test construct reliability and validity, Composite Reliability (CR) values of the constructed range between 0.755 to 0.939 which exceeds our minimum cut-off value of 0.70 [6], the average variance extracted (AVE) values were greater than 0.50, indicates that the constructs were appropriate [7], also the Cronbach's coefficient alpha score must be greater than 0.70 [8]. Finally, as can be seen in Table 5, all the constructs with their respective indicators of our study were reported to be reliable and valid for further analysis.

**3.3.3 Structural model.** To investigate whether the factor structure can be replicated in the dataset from 200 participants, a structural equation model (Fig 3) was performed after

**Table 5. Test of Reliability and validity.**

| Constructs | Items | Loading | CR | AVE | Cronbach's Alpha |
|---|---|---|---|---|---|
| Psychological health | PH1 | 0.84 | | | |
| | PH2 | 0.61 | 0.826 | 0.617 | 0.808 |
| | PH3 | 0.88 | | | |
| Work stress | WS1 | 0.67 | | | |
| | WS2 | 0.72 | 0.817 | 0.60 | 0.862 |
| | WS3 | 0.92 | | | |
| Pain severity | PS1 | 0.96 | | | |
| | PS2 | 0.88 | 0.851 | 0.879 | 0.79 |
| | PS3 | 0.54 | | | |
| | PS4 | 0.56 | | | |
| Treatment | TRT1 | 0.93 | | | |
| | TRT2 | 0.93 | 0.861 | 0.705 | 0.901 |
| | TRT3 | 0.84 | | | |
| | TRT4 | 0.62 | | | |

**Table 6. Model fit indices.**

| Goodness of fit Indices | Meaning | Value | Recommended value |
|---|---|---|---|
| $\chi^2/df$ | Relative Chi-square | 4.28 | <5.00 |
| CFI | Comparative Fit Index | 0.910 | >0.90 |
| IFI | Incremental Fit Index | 0.911 | >0.90 |
| TLI | Tucker-Lewis Index | 0.908 | >0.90 |
| PCFI | Parsimonious Comparative Fit Index | 0.690 | >0.50 |
| PNFI | Parsimonious Normed Fit Index | 0.673 | >0.50 |
| RMSEA | Root Mean Square of Error Approximation | 0.079 | <0.08 |
| SRMR | Standardized Root Mean Square Residual | 0.075 | <0.08 |

Note:$\chi^2/df$ <5.00 [9]; CFI>0.90 [10]; IFI>0.90 [11] TLI >0.90 [11]; PCFI>0.50, PNFI>0.50 [12]; RMSEA < 0.08 [13]; SRMR< 0.08 [6].

fitting CFA. There are many fit indices available to test the goodness of fit in CFA, but only a few have been used widely which are given below for our research (Table 6).

The path coefficients of the SEM identified that all null hypotheses are rejected in our model. Firstly, hypothesis $H_1$ (psychological health is positively related to the pain severity of a patient) had an acceptable strength ($\beta$ = 0.745, p <0.001) and a positive direction (Table 7). Secondly, hypothesis $H_2$ (psychological health is positively related to the treatment of a patient) shows an acceptable strength ($\beta$ = 0.401, p <0.001) and a positive direction (Table 7). Thirdly, hypothesis $H_3$ (work stress is positively related to the pain severity of a patient) had an acceptable strength ($\beta$ = 0.544, p < 0.001) and a positive direction (Table 7). Fourthly, hypothesis $H_4$ (work stress is positively related to the treatment of a patient) had an acceptable strength ($\beta$ = 0.322, p< 0.05) and a positive direction (Table 7). Lastly, hypothesis $H_5$ (pain severity is positively related to the treatment of patients) shows an acceptable strength ($\beta$ = 0.801, p < 0.001) and a positive direction (Table 7).

## 4 Summary and conclusion

The Number of Low back pain patients is increasing in our high-density populated Bangladesh country. Nowadays, people are concerned about their low back pain disease, so they go to the Center for the Rehabilitation of the Paralysed (CRP), Savar, Dhaka center to get therapy to minimize this pain. In the provided demographic data offers a snapshot of the surveyed population, revealing patterns and trends across various factors. Notably, the higher representation of males (64%) compared to females (36%) suggests a potential gender bias or variation in participation rates. The age distribution highlights a focus on the 39 to 45-year-old group (21.5%), indicating a targeted examination of the challenges faced by individuals in their prime working

**Table 7. Results of hypothesis tests based on SEM model.**

| Hypothesis | Hypothesis path | Estimate | S.E | P | Decision |
|---|---|---|---|---|---|
| $H_1$ | Psychological health →Pain severity | 0.745 | 0.123 | *** | Accepted |
| $H_2$ | Psychological health→ Treatment | 0.401 | 0.045 | *** | Accepted |
| $H_3$ | Work stress → Pain severity | 0.544 | 0.031 | *** | Accepted |
| $H_4$ | Work stress →Treatment | 0.322 | 0.057 | ** | Accepted |
| $H_5$ | Pain severity →Treatment | 0.801 | 0.069 | *** | Accepted |

P< 0.05** and P< 0.001***

years. Conversely, the limited representation of the 74 to 80-year-old age group (1%) raises questions about the inclusivity and scope of the study for older demographics. Occupationally, housewives emerge as a prominent group experiencing the effects under consideration, with 32.5% affected. This finding underscores the importance of recognizing and addressing the unique challenges faced by this demographic subset. Marital status plays a significant role, with a substantial majority being married (87%). The smaller percentages of unmarried, widowed, and divorced individuals suggest potential areas of interest for further exploration into the specific health and social dynamics within different marital statuses. Geographically, the rural-urban divide showcases a higher concentration in rural areas (59%). Understanding the implications of regional differences, including lifestyle factors, access to healthcare, and socio-economic conditions, becomes crucial in interpreting the broader context of the findings. Normal weighted people are more suffering in slight pain; on the other hand, overweight people are more suffering in moderate and severe pain. Also, the pain disability index (PDI) is significantly associated with sex, BMI, living place, and educational status.

In terms of the strength of the relationship, the SEM model revealed a strong and significant relationship between psychological health and pain severity (0.745) (Fig 3 and Table 7). That suggests that an increasing psychological health problem is directly affecting the pain severity by increasing the level of pain. This may happen due to feeling depression, fear of the panic of experiencing an attack, and how bad the pain is. We know that psychological health problems are so much dangerous for any kind of disease.

Also, psychological health had a strong and significant relationship with treatment (0.401) (Fig 3 and Table 7). It suggests that before getting treatment (Therapy) of a patient given by a physiotherapist important to know the psychological condition of the patient. Because treatment effects more effectively work if knew the mental condition of the patients.

The present study found a significant relationship between work stress and pain severity (0.544) (Fig 3 and Table 7). The possible reason leads that work stress may affect pain severity because it indirectly leads the psychological health with a positive correlation between them (0.56) (Fig 3). It means that work stress (problem with doing work, changes in the workplace, pressure in work) increases the pain severity of the patients.

In our study, the SEM model revealed another relationship between work stress and treatment (Therapy) (0.322) (Fig 3 and Table 7). After measuring the level of work stress if a physiotherapist suggests treatment (therapy) for a patient, that results in the patient reducing the pain level of the lower back but it is not the right way to get treatment without measuring the psychological health condition and pain severity. Physiotherapists did not suggest this way of getting treatment.

The most important relationship revealed by the SEM model in our present study is a strong and positive relationship between pain severity and treatment (0.801) (Fig 3 and Table 7). It suggests that treatment has a direct effect on pain severity. After getting treatment, the pain severity decreases day by day and it is evident from the factor loadings of treatment with their observed variables (Day wise treatment protocol). By observing the factor loading of treatment variable, we also conclude that first-day treatment is more significant than another day, second-day treatment is less significant than the first day, and the significance level declines gradually day by day.

In conclusion, the observation provides valuable insights into the complex interplay between demographic factors, psychological health, work stress, and the effectiveness of treatment in managing low back pain. This information is crucial for developing targeted interventions and strategies for improving the overall well-being of individuals experiencing low back pain in Bangladesh.

## 5 Limitation of the study

While collecting data in CRP, some of the patients have treated interviewer as interns or representatives of the physician and were reluctant to pay attention to their several problems. Although some of the respondents were cooperative in providing the information during data collection, some were reluctant; some were shy in providing accurate information. Besides, some could not provide information at all due to unbearable pain.

Also, the survey was performed within a short time and within budget limits. However, despite this limitation, the present study is expected to satisfy all the objectives that it claims.

## Supporting information

**S1 File. Survey questionnaire, permission letter and dataset of this study.**
(ZIP)

## Acknowledgments

The authors wish to thank healthcare technicians of CRP, Savar for their assistance and show gratitude to all the participants of this study for their willingness and active participation.

## Author Contributions

**Conceptualization:** Md. Forhad Hossain.

**Data curation:** Mohammad Arifur Rahman.

**Formal analysis:** Sanjida Tasnim.

**Methodology:** Sanjida Tasnim.

**Software:** Mohammad Arifur Rahman, Sanjida Tasnim.

**Supervision:** Md. Forhad Hossain.

**Visualization:** Mohammad Arifur Rahman.

**Writing – original draft:** Mohammad Arifur Rahman.

**Writing – review & editing:** Sanjida Tasnim.

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
