## [Decision Letter · Decision Letter 0]

3 Jan 2024

PONE-D-23-26883A Structural Equation Modeling to Determine the Pain Severity and Treatment Effect of the Patients with Low Back Pain Attending at Center for the Rehabilitation of the Paralysed, BangladeshPLOS ONE

Dear Dr. Tasnim,

Thank you for submitting your manuscript to PLOS ONE. After careful consideration, we feel that it has merit but does not fully meet PLOS ONE’s publication criteria as it currently stands. Therefore, we invite you to submit a revised version of the manuscript that addresses the points raised during the review process. Please submit your revised manuscript by Feb 17 2024 11:59PM. If you will need more time than this to complete your revisions, please reply to this message or contact the journal office at plosone@plos.org. Please include the following items when submitting your revised manuscript:A rebuttal letter that responds to each point raised by the academic editor and reviewer(s). You should upload this letter as a separate file labeled 'Response to Reviewers'.A marked-up copy of your manuscript that highlights changes made to the original version. You should upload this as a separate file labeled 'Revised Manuscript with Track Changes'.An unmarked version of your revised paper without tracked changes. You should upload this as a separate file labeled 'Manuscript'.

We look forward to receiving your revised manuscript.

Kind regards,

Md. Feroz Kabir, BPT, MPT, MPH, BPED, MPED

Academic Editor

PLOS ONE

Journal Requirements:

3. We note you have included a table to which you do not refer in the text of your manuscript. Please ensure that you refer to Tables 4 and 6 in your text; if accepted, production will need this reference to link the reader to the Table.

Additional Editor Comments:

Please rectify the following things:.

Rahman et al. present a study of factors influencing the severity of lower back pain. This is an important topic. The authors managed to collect a reasonably sized dataset in a relatively short amount of time, which is commendable. However, I have several concerns about the analysis and its presentation.

General Comments

The quality of written English is insufficient for publication. There are frequent unusual phrasings and ungrammatical sentences. Sometimes to the extent that the intended meaning becomes difficult to discern. There are too many instances to list, but I’ve provided some examples in the detailed comments below. I would strongly suggest that the authors obtain help editing the manuscript, ideally from a native speaker.

Detailed Comments

Major Comments

1. The presentation of the data collection and analysis is missing important details.

a. Where did the questionnaire used to collect the data for this study originate? Was that developed previously, or was it developed as part of this study? Based on what is presented in the manuscript I suspect the latter. This should be clarified, and if the questionnaire was indeed developed for this study more details on the process used should be provided. How were questions developed and what was done to ensure face validity and content validity?

b. How was the association of individual items and constructs established? There is no mention of an exploratory factor analysis. Was this done?

c. There seems to be very high correlation between Work Stress, Psychological Health, and Pain Severity, suggesting that the factor structure could be improved. Did the authors examine cross-loadings of the items used?

d. It appears that the same dataset may have been used for exploratory and confirmatory factor analysis (if exploratory factor analysis was carried out at all). A rigorous analysis of this type would require that the questionnaire be assessed for validity, exploratory factor analysis is carried out to establish the factor structure (possibly refining the questionnaire to deal with cross-loadings and otherwise problematic items), a new, independent sample is collected to carry out confirmatory factor analysis to establish that the factor structure is appropriate and robust. Then SEM can be used for a more detailed analysis. Much of this is missing from the description in the manuscript. The authors should clarify to what extent the various stages of the process have taken place.

2. It is unclear to me how treatment was coded. It appears that treatment was recorded for four days, and the manuscript claims that the analysis shows that pain changed in response to treatment, but the model does not seem to model change over time. There is no indication that pain severity was recorded more than once.

3. The authors make several claims that are not supported by their data:

a. The claim that “The age group of 39 to 45 years people is more affected by low back pain” (p. 4, lines 70 -71) cannot be supported by the data presented here. The statement implies that 39 – 45-year-olds are more likely to present with LBP. To establish this, a comparison between the age distribution of the target population (everyone who would visit the centre if they were suffering from LBP) and the observed age distribution of patients. This is not possible based on the patient data alone. Similar considerations apply to the other factors examined in the univariate analyses. The authors should be careful how they present these results. The univariate analyses are useful in characterising the sample, but no inference can be drawn from them.

b. Similarly, the claims on p. 8, lines 139 – 141, seem to rely solely on the frequency with which different groups were observed in the sample. Without reference to an appropriate comparator these do not allow any inference about the relative frequency of LBP.

c. The claim that male participants are more affected by LBP than females because they are “less concerned about their health issues than females” (p. 8, lines 137-138) is simply speculation and not supported by the data. While it is valuable to speculate about the reasons underlying the findings of a study to generate hypotheses for future research, they should be presented as such.

4. I’m not convinced by the bivariate analyses. The authors do not explain what analyses were carried out to obtain the p values. There is also no indication whether the reported p values were adjusted for multiple testing. More importantly, If these factors indeed all contribute to pain severity, they should not be considered in isolation. If the authors wish to carry out a cross-sectional analysis like this, a multiple regression model seems more appropriate.

5. When reporting the outcome of statistical tests, specific p values should be reported. It is not sufficient to state p < 0.05.

6. When reporting the outcome of hypothesis tests, the authors say that “the hypothesis was accepted” when they mean that the null hypothesis was rejected. This wording should be changed to avoid confusion.

Minor Comments

7. The abstract states that “The majority of the respondents were married 174(87%), followed by unmarried respondents …”. It is unclear to me that the distribution of marital status in the sample is sufficiently relevant to merit inclusion in the abstract.

8. Table 1 has a column labelled “Latent constructs”. These appear to be short labels for the individual items, not latent constructs.

9. The item descriptions in Table 1 should be re-worded to better conform with standard usage. For example, “pain of badness” should probably be ‘pain intensity’(I think. It is possible that something else is meant here).

10. Table 3 is labelled “Test of reliability and validity” but appears to show results of the bivariate analyses.

11. Typographical and grammatical errors

a. Abstract, Study design: “… the department of the physiotherapy department …” should be either “the department of physiotherapy” or “the physiotherapy department”.

b. Abstract, Conclusion: “Also pain severity of the patients is significantly associated in treatment scheme with strong strength.” Maybe something like “Pain severity is significantly associated with treatment scheme intensity” would work better here. I’m unsure what the different treatment schemes are and whether ‘intensity’ appropriately describes their difference.

c. p. 2, line 7: “subjects experience mild, severe or acute pair which can last from few days to few years” The use of ‘acute’ in this sentence may be inappropriate. Unlike ‘mild’ and ‘severe’, which indicate different levels of severity, it is typically used to indicate sudden and short periods (of often severe) pain, as opposed to chronic pain.

d. P 2. Line 8: “almost 12% and 44% have LBP at any particular time” This should be something like “12% to 44% have LBP at any particular time”.

e. p. 2, lines 12-14: “Some demographic features such as age, gender etc and other some known risk factors for LBP.” Dele the second instance of “some”.

f. P. 2, line 15: “lumber lordships”. I’m not sure what this is supposed to be. My best guess is ‘lumbar lordosis’.

Reviewers' comments:

Reviewer's Responses to Questions

**Comments to the Author**

1. Is the manuscript technically sound, and do the data support the conclusions?

Reviewer #1: No

Reviewer #2: No

2. Has the statistical analysis been performed appropriately and rigorously? 

Reviewer #1: No

Reviewer #2: No

3. Have the authors made all data underlying the findings in their manuscript fully available?

Reviewer #1: No

Reviewer #2: Yes

4. Is the manuscript presented in an intelligible fashion and written in standard English?

Reviewer #1: No

Reviewer #2: No

5. Review Comments to the Author

Reviewer #1: Thank you for letting me review this manuscript. I have some comments that need attention before any decision can be made.

ABSTRACT

The abstract should never have reference on tables or on studies. The description of your results lacks objectivity, and you should rewrite it for clarity.

You use abbreviations (BMI, SEM) that were not previously defined.

The conclusion should be more “straight to the point”, and should only focus on the findings of your study.

The keywords should be more of MeSH terms and less on the statistical analysis you performed.

INTRODUCTION

Lines 6-7: not all LBP comes from injuries; acute pain is not severe pain, please rephrase this sentence.

The introduction should be a part of the text in which you describe the problem and the need for your study. However, the need to assess socioeconomic situation, and etc., is not clear in the text. You should rewrite this whole section in order to improve readability and to really focus on what is the novelty of your study – what it really adds to the literature that was not yet studied by other groups. Invest time in this section to lead the reader to the point you are studying.

Most important than that, though, is that your title is about a modelling proposal, but your aim was to assess the effects of LBP treatments to reduce pain. As such, you need to either change your title, or rewrite your aims (which I believe is the case). In the abstract, you comment about several hypothesis, and they should be clearly stated here, right after the aims of the study. And by hypothesis, I mean the study hypothesis, and not the statistical hypothesis you have elaborated.

MATERIALS AND METHODS

You study lacks description of the instruments used to assess pain severity, low back pain index, work condition, work pressure and etc.. You need to describe thoroughly the instruments you used. If you only used open questions or if you developed an instrument, you should describe it so that readers can understand what was actually assessed by you.

The eligibility criteria are not clear: did you assess men and women? How old should they be to be included or excluded? Did you assess if pain was really from a mechanical origin? How was it assessed? How many days of pain they reported? Did you exclude people that reported chronic LBP? What was your criteria of treatment? Any type of treatment was included? How long have they been undergoing treatment? Please add a complete description of the eligibility criteria.

Why a minimum of 4 days of treatment was established? How did you collect data? In person, through an electronic form? Did you perform physical assessments? Please describe better those methods, especially the instruments you used.

RESULTS

You included people from 18 to 87, which I do not think is adequate. Over 20% of your sample is ≥60 years old, and their data should be very different from people aged 18-29 and from people aged 30-59. You need to rethink your inclusion criteria.

What is pain disability index? What is this instrument?

You present a model for the analysis of an instrument, yet, you do not present a single line on this process.

DISCUSSION

You need to discuss your results, and after reading the paper, I do not know if you created a framework to assess treatment effects on people with acute LBP, if you assessed people with acute LBP and checked if physical therapy worked for them, if you checked whether work and education influence on the effects of physical therapy for LBP or if you created an instrument to understand how other factors influence on LBP.

Reviewer #2: Rahman et al. present a study of factors influencing the severity of lower back pain. This is an important topic. The authors managed to collect a reasonably sized dataset in a relatively short amount of time, which is commendable. However, I have several concerns about the analysis and its presentation.

General Comments

The quality of written English is insufficient for publication. There are frequent unusual phrasings and ungrammatical sentences. Sometimes to the extent that the intended meaning becomes difficult to discern. There are too many instances to list, but I’ve provided some examples in the detailed comments below. I would strongly suggest that the authors obtain help editing the manuscript, ideally from a native speaker.

Detailed Comments

Major Comments

1. The presentation of the data collection and analysis is missing important details.

a. Where did the questionnaire used to collect the data for this study originate? Was that developed previously, or was it developed as part of this study? Based on what is presented in the manuscript I suspect the latter. This should be clarified, and if the questionnaire was indeed developed for this study more details on the process used should be provided. How were questions developed and what was done to ensure face validity and content validity?

b. How was the association of individual items and constructs established? There is no mention of an exploratory factor analysis. Was this done?

c. There seems to be very high correlation between Work Stress, Psychological Health, and Pain Severity, suggesting that the factor structure could be improved. Did the authors examine cross-loadings of the items used?

d. It appears that the same dataset may have been used for exploratory and confirmatory factor analysis (if exploratory factor analysis was carried out at all). A rigorous analysis of this type would require that the questionnaire be assessed for validity, exploratory factor analysis is carried out to establish the factor structure (possibly refining the questionnaire to deal with cross-loadings and otherwise problematic items), a new, independent sample is collected to carry out confirmatory factor analysis to establish that the factor structure is appropriate and robust. Then SEM can be used for a more detailed analysis. Much of this is missing from the description in the manuscript. The authors should clarify to what extent the various stages of the process have taken place.

2. It is unclear to me how treatment was coded. It appears that treatment was recorded for four days, and the manuscript claims that the analysis shows that pain changed in response to treatment, but the model does not seem to model change over time. There is no indication that pain severity was recorded more than once.

3. The authors make several claims that are not supported by their data:

a. The claim that “The age group of 39 to 45 years people is more affected by low back pain” (p. 4, lines 70 -71) cannot be supported by the data presented here. The statement implies that 39 – 45-year-olds are more likely to present with LBP. To establish this, a comparison between the age distribution of the target population (everyone who would visit the centre if they were suffering from LBP) and the observed age distribution of patients. This is not possible based on the patient data alone. Similar considerations apply to the other factors examined in the univariate analyses. The authors should be careful how they present these results. The univariate analyses are useful in characterising the sample, but no inference can be drawn from them.

b. Similarly, the claims on p. 8, lines 139 – 141, seem to rely solely on the frequency with which different groups were observed in the sample. Without reference to an appropriate comparator these do not allow any inference about the relative frequency of LBP.

c. The claim that male participants are more affected by LBP than females because they are “less concerned about their health issues than females” (p. 8, lines 137-138) is simply speculation and not supported by the data. While it is valuable to speculate about the reasons underlying the findings of a study to generate hypotheses for future research, they should be presented as such.

4. I’m not convinced by the bivariate analyses. The authors do not explain what analyses were carried out to obtain the p values. There is also no indication whether the reported p values were adjusted for multiple testing. More importantly, If these factors indeed all contribute to pain severity, they should not be considered in isolation. If the authors wish to carry out a cross-sectional analysis like this, a multiple regression model seems more appropriate.

5. When reporting the outcome of statistical tests, specific p values should be reported. It is not sufficient to state p < 0.05.

6. When reporting the outcome of hypothesis tests, the authors say that “the hypothesis was accepted” when they mean that the null hypothesis was rejected. This wording should be changed to avoid confusion.

Minor Comments

7. The abstract states that “The majority of the respondents were married 174(87%), followed by unmarried respondents …”. It is unclear to me that the distribution of marital status in the sample is sufficiently relevant to merit inclusion in the abstract.

8. Table 1 has a column labelled “Latent constructs”. These appear to be short labels for the individual items, not latent constructs.

9. The item descriptions in Table 1 should be re-worded to better conform with standard usage. For example, “pain of badness” should probably be ‘pain intensity’(I think. It is possible that something else is meant here).

10. Table 3 is labelled “Test of reliability and validity” but appears to show results of the bivariate analyses.

11. Typographical and grammatical errors

a. Abstract, Study design: “… the department of the physiotherapy department …” should be either “the department of physiotherapy” or “the physiotherapy department”.

b. Abstract, Conclusion: “Also pain severity of the patients is significantly associated in treatment scheme with strong strength.” Maybe something like “Pain severity is significantly associated with treatment scheme intensity” would work better here. I’m unsure what the different treatment schemes are and whether ‘intensity’ appropriately describes their difference.

c. p. 2, line 7: “subjects experience mild, severe or acute pair which can last from few days to few years” The use of ‘acute’ in this sentence may be inappropriate. Unlike ‘mild’ and ‘severe’, which indicate different levels of severity, it is typically used to indicate sudden and short periods (of often severe) pain, as opposed to chronic pain.

d. P 2. Line 8: “almost 12% and 44% have LBP at any particular time” This should be something like “12% to 44% have LBP at any particular time”.

e. p. 2, lines 12-14: “Some demographic features such as age, gender etc and other some known risk factors for LBP.” Dele the second instance of “some”.

f. P. 2, line 15: “lumber lordships”. I’m not sure what this is supposed to be. My best guess is ‘lumbar lordosis’.

6. PLOS authors have the option to publish the peer review history of their article (what does this mean?). If published, this will include your full peer review and any attached files.

Reviewer #1: No

Reviewer #2: **Yes: **Peter Humburg

While revising your submission, please upload your figure files to the Preflight Analysis and Conversion Engine (PACE) digital diagnostic tool, https://pacev2.apexcovantage.com/. PACE helps ensure that figures meet PLOS requirements. To use PACE, you must first register as a user. Registration is free. Then, login and navigate to the UPLOAD tab, where you will find detailed instructions on how to use the tool. If you encounter any issues or have any questions when using PACE, please email PLOS at figures@plos.org. Please note that supporting Information files do not need this step.

---

## [Author Response · Author response to Decision Letter 0]

20 Mar 2024

Author responses to the review comments:

We would like to express our sincere gratitude to the two reviewers and the Academic Editor for their valuable comments. We have considered all the comments made by the reviewers and thoroughly revised and formatted the manuscript accordingly. A detailed response to each of the comments is provided in the table below:

Academic Editor comments: Response Note

Manuscript Number PONE-D-23-26883 entitled “A Structural Equation Modeling to Determine the Pain Severity and Treatment Effect of the Patients with Low Back Pain Attending at Center for the Rehabilitation of the Paralysed, Bangladesh" which you submitted to PLOS ONE, has been reviewed. The comments of the reviewer(s) are included at the bottom of this letter.

 The reviewer(s) suggest some revisions to your manuscript before it can be considered for publication. Therefore, I invite you to respond to the reviewer(s)' comments and revise your manuscript. Thank you very much for completing the review process of our manuscript. We are thankful for providing the feedback which is helpful to improve the overall quality of our manuscript. We have substantially revised our manuscript by addressing all the reviewer’s comments which we would like to submit again to your journal. 

We upload the revised versions (track change and clean) to the journal system. Revisions are in red color. 

Reviewer 1 comments: Response Note

ABSTRACT

The abstract should never have reference on tables or on studies. The description of your results lacks objectivity, and you should rewrite it for clarity.

You use abbreviations (BMI, SEM) that were not previously defined.

The conclusion should be more “straight to the point”, and should only focus on the findings of your study.

The keywords should be more of MeSH terms and less on the statistical analysis you performed. Thank you very much for your comments. We revised the manuscript as per your comments. Revisions are in red color. 

Page: 1

 INTRODUCTION

Lines 6-7: not all LBP comes from injuries; acute pain is not severe pain, please rephrase this sentence.

The introduction should be a part of the text in which you describe the problem and the need for your study. However, the need to assess socioeconomic situation, and etc., is not clear in the text. You should rewrite this whole section in order to improve readability and to really focus on what is the novelty of your study – what it really adds to the literature that was not yet studied by other groups. Invest time in this section to lead the reader to the point you are studying.

Most important than that, though, is that your title is about a modelling proposal, but your aim was to assess the effects of LBP treatments to reduce pain. As such, you need to either change your title, or rewrite your aims (which I believe is the case). In the abstract, you comment about several hypothesis, and they should be clearly stated here, right after the aims of the study. And by hypothesis, I mean the study hypothesis, and not the statistical hypothesis you have elaborated. 

Thank you very much for your careful checking. We revised it. 

Revisions are in red color. 

Page: 02-03

MATERIALS AND METHODS

You study lacks description of the instruments used to assess pain severity, low back pain index, work condition, work pressure and etc. You need to describe thoroughly the instruments you used. If you only used open questions or if you developed an instrument, you should describe it so that readers can understand what was actually assessed by you.

The eligibility criteria are not clear: did you assess men and women? How old should they be to be included or excluded? Did you assess if pain was really from a mechanical origin? How was it assessed? How many days of pain they reported? Did you exclude people that reported chronic LBP? What was your criteria of treatment? Any type of treatment was included? How long have they been undergoing treatment? Please add a complete description of the eligibility criteria.

Why a minimum of 4 days of treatment was established? How did you collect data? In person, through an electronic form? Did you perform physical assessments? Please describe better those methods, especially the instruments you used. 

Thank you very much. We revised and added more information.

Men and women both are assessed. There was no age barrier for being included in the study. Each subject was asked to report the duration of pain problem. Each subject is assessed with an initial and outcome level of pain intensity after receiving certain doses of specific or need based treatment which the physiotherapist suggested for him/her. 

4 days treatment period was set with the suggestion of physiotherapist as patient begins to show good response after first four sessions. The data was collected in person through a paper- based questionnaire. 

Revisions are in red color. 

Page: 04

RESULTS

You included people from 18 to 87, which I do not think is adequate. Over 20% of your sample is ≥60 years old, and their data should be very different from people aged 18-29 and from people aged 30-59. You need to rethink your inclusion criteria.

What is pain disability index? What is this instrument?

You present a model for the analysis of an instrument, yet, you do not present a single line on this process. 

There was no age barrier for being included in the study. Rather than studying the age effect on pain, the study focuses on identifying the inter relation of phycological health, work, pain severity and treatment effect. 

Revisions are in red color. 

Page: 03

DISCUSSION

You need to discuss your results, and after reading the paper, I do not know if you created a framework to assess treatment effects on people with acute LBP, if you assessed people with acute LBP and checked if physical therapy worked for them, if you checked whether work and education influence on the effects of physical therapy for LBP or if you created an instrument to understand how other factors influence on LBP. 

Many thanks. The authors have checked and revised the whole manuscript and broadly discussed the results. 

Revisions are in red color. 

Page: 06-13

Reviewer 2 comments: Response Note

General Comments 

The quality of written English is insufficient for publication. There are frequent unusual phrasings and ungrammatical sentences. Sometimes to the extent that the intended meaning becomes difficult to discern. There are too many instances to list, but I’ve provided some examples in the detailed comments below. I would strongly suggest that the authors obtain help editing the manuscript, ideally from a native speaker Thank you very much for your comments and feedback. We believe that this help to improve the quality of the manuscript. 

 Revisions are in red color. 

 Major Comments

 Response Note

1. The presentation of the data collection and analysis is missing important details.

a. Where did the questionnaire used to collect the data for this study originate? Was that developed previously, or was it developed as part of this study? Based on what is presented in the manuscript I suspect the latter. This should be clarified, and if the questionnaire was indeed developed for this study more details on the process used should be provided. How were questions developed and what was done to ensure face validity and content validity?

b. How was the association of individual items and constructs established? There is no mention of an exploratory factor analysis. Was this done?

c. There seems to be very high correlation between Work Stress, Psychological Health, and Pain Severity, suggesting that the factor structure could be improved. Did the authors examine cross-loadings of the items used?

d. It appears that the same dataset may have been used for exploratory and confirmatory factor analysis (if exploratory factor analysis was carried out at all). A rigorous analysis of this type would require that the questionnaire be assessed for validity, exploratory factor analysis is carried out to establish the factor structure (possibly refining the questionnaire to deal with cross-loadings and otherwise problematic items), a new, independent sample is collected to carry out confirmatory factor analysis to establish that the factor structure is appropriate and robust. Then SEM can be used for a more detailed analysis. Much of this is missing from the description in the manuscript. The authors should clarify to what extent the various stages of the process have taken place. Many Thanks. The author has checked, added some material, and clarified the manuscript.

The questionnaire used Dallas Pain questionnaire to measure pain severity and and Oswestry low back pain disability index to measure pain disability index (PDI). These tools are valid and popular for the assessment of patients with low back pain. 

As it was a pre tested questionnaire, exploratory factor analysis (EFA) is an option, not a mandatory task. So we have excluded it from the analysis.

Cross loadings have been included. Revisions are in red color. 

Page: 03-11

2. It is unclear to me how treatment was coded. It appears that treatment was recorded for four days, and the manuscript claims that the analysis shows that pain changed in response to treatment, but the model does not seem to model change over time. There is no indication that pain severity was recorded more than once. Thank you so much and it is added in table 1. Revisions are in red color. 

Page: 05-06 

3. The authors make several claims that are not supported by their data:

a. The claim that “The age group of 39 to 45 years people is more affected by low back pain” (p. 4, lines 70 -71) cannot be supported by the data presented here. The statement implies that 39 – 45-year-olds are more likely to present with LBP. To establish this, a comparison between the age distribution of the target population (everyone who would visit the centre if they were suffering from LBP) and the observed age distribution of patients. This is not possible based on the patient data alone. Similar considerations apply to the other factors examined in the univariate analyses. The authors should be careful how they present these results. The univariate analyses are useful in characterising the sample, but no inference can be drawn from them.

b. Similarly, the claims on p. 8, lines 139 – 141, seem to rely solely on the frequency with which different groups were observed in the sample. Without reference to an appropriate comparator these do not allow any inference about the relative frequency of LBP.

c. The claim that male participants are more affected by LBP than females because they are “less concerned about their health issues than females” (p. 8, lines 137-138) is simply speculation and not supported by the data. While it is valuable to speculate about the reasons underlying the findings of a study to generate hypotheses for future research, they should be presented as such. Thank you very much. The author has checked and revised the manuscript.

Thank you very much. The authors have checked and revised the manuscript.

Thank you very much. The authors have checked and revised the manuscript.

 Revisions are in red color. 

Page: 06-07

4.I’m not convinced by the bivariate analyses. The authors do not explain what analyses were carried out to obtain the p values. There is also no indication whether the reported p values were adjusted for multiple testing. More importantly, if these factors indeed all contribute to pain severity, they should not be considered in isolation. If the authors wish to carry out a cross-sectional analysis like this, a multiple regression model seems more appropriate. We truly appreciate this valuable suggestion. P value comes from Chi square test which shows the association between the variables. 

As demographical factors are not the main contributor to pain severity, a multiple regression has not been performed. Revisions are in red color. 

Page: 08

5. When reporting the outcome of statistical tests, specific p values should be reported. It is not sufficient to state p < 0.05. Thank you so much. Revisions are in red color. 

Page: 09

6. When reporting the outcome of hypothesis tests, the authors say that “the hypothesis was accepted” when they mean that the null hypothesis was rejected. This wording should be changed to avoid confusion. Many thanks and it is revised in the manuscript. Revisions are in red color. 

Page: 13

Minor Comments Response Note

7. The abstract states that “The majority of the respondents were married 174(87%), followed by unmarried respondents …”. It is unclear to me that the distribution of marital status in the sample is sufficiently relevant to merit inclusion in the abstract.

 Thank you so much and it has been excluded. 

8. Table 1 has a column labelled “Latent constructs”. These appear to be short labels for the individual items, not latent constructs.

 Thank you. Revisions are in red color. 

Page: 05

9. The item descriptions in Table 1 should be re-worded to better conform with standard usage. For example, “pain of badness” should probably be ‘pain intensity’ (I think. It is possible that something else is meant here).

 Thank you. Revisions are in red color. 

Page: 05

10. Table 3 is labelled “Test of reliability and validity” but appears to show results of the bivariate analyses.

 Thank you so much. Revisions are in red color. 

Page: 08

11. Typographical and grammatical errors

a. Abstract, Study design: “… the department of the physiotherapy department …” should be either “the department of physiotherapy” or “the physiotherapy department”.

b. Abstract, Conclusion: “Also pain severity of the patients is significantly associated in treatment scheme with strong strength.” Maybe something like “Pain severity is significantly associated with treatment scheme intensity” would work better here. I’m unsure what the different treatment schemes are and whether ‘intensity’ appropriately describes their difference.

c. p. 2, line 7: “subjects experience mild, severe or acute pair which can last from few days to few years” The use of ‘acute’ in this sentence may be inappropriate. Unlike ‘mild’ and ‘severe’, which indicate different levels of severity, it is typically used to indicate sudden and short periods (of often severe) pain, as opposed to chronic pain.

d. P 2. Line 8: “almost 12% and 44% have LBP at any particular time” This should be something like “12% to 44% have LBP at any particular time”.

e. p. 2, lines 12-14: “Some demographic features such as age, gender etc and other some known risk factors for LBP.” Dele the second instance of “some”.

f. P. 2, line 15: “lumber lordships”. I’m not sure what this is supposed to be. My best guess is ‘lumbar lordosis’. We truly appreciate this valuable suggestion. We have revised all of these. Revisions are in red color. 

Page: 01-02

Finally, the revised manuscript has been produced following the valuable comments and suggestions of the reviewers. Once again, we would like to thank the reviewers for their sincere dedication, professional insights, and earnest cooperation in reviewing the manuscript.

---

## [Editor Report · Decision Letter 1]

3 May 2024

Assessing Pain Severity and Treatment Outcomes in Patients with Low Back Pain: A Structural Equation Modeling Approach at the Center for the Rehabilitation of the Paralysed, Bangladesh

PONE-D-23-26883R1

Dear Sanjida Tasnim,

We’re pleased to inform you that your manuscript has been judged scientifically suitable for publication and will be formally accepted for publication once it meets all outstanding technical requirements.

Kind regards,

Md. Feroz Kabir, BPT, MPT, MPH, BPED, MPED

Academic Editor

PLOS ONE

Additional Editor Comments (optional):

Dear Authors, Your revised manuscript is getting consideration for acceptance, but you need to improve the English grammar thoroughly and submit it within the next 15 days.
---

## [Editor Report · Acceptance letter]

21 May 2024

PONE-D-23-26883R1 

PLOS ONE

Dear Dr. Tasnim, 

I'm pleased to inform you that your manuscript has been deemed suitable for publication in PLOS ONE. Congratulations! Your manuscript is now being handed over to our production team.

Kind regards, 

on behalf of

Dr. Md. Feroz Kabir 

Academic Editor

PLOS ONE